# Ta$_2$O$_5$/SiO$_2$ Multicomponent Dielectrics for Amorphous Oxide TFTs

**Jorge Martins** [1,*] , **Asal Kiazadeh** [1], **Joana V. Pinto** [1], **Ana Rovisco** [1], **Tiago Gonçalves** [1], **Jonas Deuermeier** [1],
**Eduardo Alves** [2], **Rodrigo Martins** [1], **Elvira Fortunato** [1] **and Pedro Barquinha** [1,*]

[1]  i3N/CENIMAT, Department of Materials Science, NOVA School of Science and Technology and
    CEMOP/UNINOVA, NOVA University Lisbon, Campus de Caparica, 2829-516 Caparica, Portugal;
    a.kiazadeh@fct.unl.pt (A.K.); jdvp@fct.unl.pt (J.V.P.); a.rovisco@campus.fct.unl.pt (A.R.);
    td.goncalves@campus.fct.unl.pt (T.G.); j.deuermeier@campus.fct.unl.pt (J.D.); rfpm@fct.unl.pt (R.M.);
    emf@fct.unl.pt (E.F.)
[2]  IPFN, Instituto Superior Técnico, University of Lisbon, EN 10, km 139,7 2695-066 Bobadela, Portugal;
    ealves@ctn.tecnico.ulisboa.pt
[*]  Correspondence: jds.martins@campus.fct.unl.pt (J.M.); pmcb@fct.unl.pt (P.B.)

**Abstract:** Co-sputtering of SiO$_2$ and high-κ Ta$_2$O$_5$ was used to make multicomponent gate dielectric stacks for In-Ga-Zn-O thin-film transistors (IGZO TFTs) under an overall low thermal budget (T = 150 °C). Characterization of the multicomponent layers and of the TFTs working characteristics (employing them) was performed in terms of static performance, reliability, and stability to understand the role of the incorporation of the high-κ material in the gate dielectric stack. It is shown that inherent disadvantages of the high-κ material, such as poorer interface properties and poor gate insulation, can be counterbalanced by inclusion of SiO$_2$ both mixed with Ta$_2$O$_5$ and as thin interfacial layers. A stack comprising a (Ta$_2$O$_5$)$_x$(SiO$_2$)$_{100-x}$ film with x = 69 and a thin SiO$_2$ film at the interface with IGZO resulted in the best performing TFTs, with field-effect mobility (μ$_{FE}$ ≈ 16 cm$^2$·V$^{-1}$·s$^{-1}$), subthreshold slope (SS) ≈ 0.15 V/dec and on/off ratio exceeding 10$^7$. Anomalous $V_{th}$ shifts were observed during positive gate bias stress (PGBS), followed by very slow recoveries (time constant exceeding 8 × 10$^5$ s), and analysis of the stress and recovery processes for the different gate dielectric stacks showed that the relevant mechanism is not dominated by the interfaces but seems to be related to the migration of charged species in the dielectric. The incorporation of additional SiO$_2$ layers into the gate dielectric stack is shown to effectively counterbalance this anomalous shift. This multilayered gate dielectric stack approach is in line with both the large area and the flexible electronics needs, yielding reliable devices with performance suitable for successful integration on new electronic applications.

**Keywords:** Ta$_2$O$_5$/SiO$_2$; TFTs; anomalous $V_{th}$ shift; multicomponent dielectrics; high-κ dielectrics

## 1. Introduction

Amorphous oxide (AO) thin films have greatly progressed in a relatively short time, having found market application in the display industry where materials such as indium-gallium-zinc oxide (IGZO) appear as an advantageous alternative to Si technologies [1,2]. Besides conventional electronics, their characteristics make them suitable for concepts such as transparent and flexible electronics [3–5] or even paper electronics [6–9], allowing for interesting applications in fields such as medical, security and item tracking [10,11], crucial under the scope of the Internet of Things (IoT). One of the main advantages of AO is their good properties even when fabricated at low temperatures, with temperatures below 200 °C being imposed when considering flexible substrates or even paper substrates. Lower annealing temperatures unavoidably result in poorer device performance and stability. When considering these lower annealing temperatures the IGZO properties are known to be strongly related to its processing conditions [12] and adjustment of the cation ratio

in the material thus plays a major role in its optimization [13]. In addition, employing dielectrics with high dielectric permittivity, $\varepsilon_r$, (high-κ dielectrics) can compensate for poorer performances by reducing driving voltages (e.g., as required in power-efficient applications within IoT) and improving gate voltage swing due to higher gate capacitances [14,15]. For low temperature deposition of dielectrics, physical techniques such as pulsed laser deposition (PLD) [16,17], thermal evaporation [18,19] and sputtering can be used, the latter allowing the deposition of most materials without any intentional substrate heating [20], at a large scale and with low contamination [21]. Several high-κ materials have been employed for gate dielectrics in low-temperature IGZO TFTs (or other ZnO-based TFTs) including: $Al_2O_3$ [22–24], $HfO_2$ [25], $Ta_2O_5$ [26,27], $Y_2O_3$ [28–30] and $ZrO_2$ [31]. Nevertheless, high-κ materials present some disadvantages, aggravated by low thermal budgets, when compared to conventional dielectrics such as $SiO_2$. While having a relatively low permittivity ($\varepsilon_r$ = 3.9), $SiO_2$ is stable, has a very high band gap ($E_g$) of 9 eV and has a low defect density making it a good insulator with a high breakdown voltage. Additionally, it is amorphous and has a good interface with IGZO. On the other hand, ionic bonds in high-κ dielectrics result in high defect concentrations with oxygen vacancies ($V_O$) being the primary source of traps. These can be a source of fixed charges or act as electron traps, scattering carriers in the channel (decreasing mobility), changing the threshold voltage ($V_{th}$) and assisting oxide breakdown and gate leakage mechanisms [32], decreasing device performance, stability and reliability. Furthermore, high-κ materials are often polycrystalline (even at low temperatures) with grain boundaries contributing both to degraded surface properties and acting as preferential paths for leakage current and impurity diffusion [33,34]. When choosing the dielectric material, band alignment should be considered as at least 1 eV of conduction/valence band offset is desirable for blocking electron/hole injection. $\varepsilon_r$ is normally inversely proportional to $E_g$ (Figure 1), and the band alignment of several dielectrics with IGZO can be found in the work of Hays et al. [35].

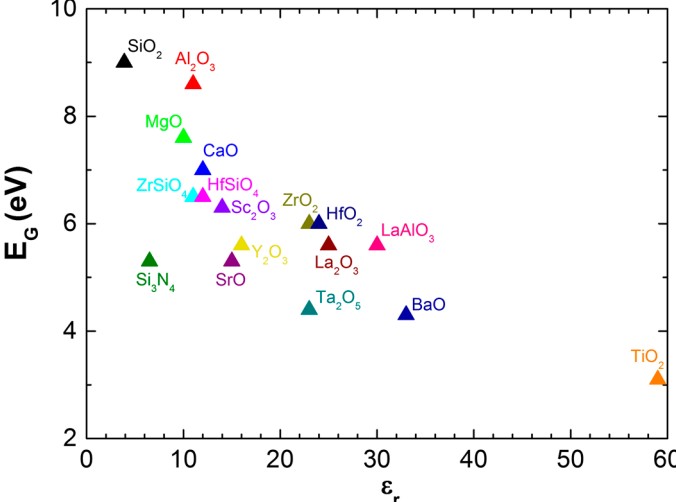

**Figure 1.** Dielectric constant versus band gap for oxides. Adapted from [35], with the permission of AIP Publishing.

Incorporation of higher $E_g$ materials with high-κ dielectrics effectively increases $E_G$ and can result in amorphous materials to much higher temperatures due to their increased disorder [36]. Regarding low temperature (T < 200 °C) ZnO-based TFTs with multicomponent dielectrics, sputtered $Bi_{1.5}Zn_{1.0}Nb_{1.5}O_7$ showed $\varepsilon_r \approx 51$ but was polycrystalline at room-temperature (RT) [37] while sputtered $Ba_{0.5}Sr_{0.5}TiO_3$ ($\varepsilon_r \approx 28$) was amorphous at RT but presented significant leakage [38]. The insertion of MgO into the latter dielectric improved insulation at the expense of $\varepsilon_r$ (to close to 18) and it was applied to IGZO TFTs with good performance on plastic substrates [39]. In previous studies conducted by our group, sputtered $HfO_2$ was combined with $SiO_2$ or $AlO_x$ effectively preventing the crystallization

of the material at RT [20,40]. Similarly, $Ta_2O_5$ was combined with $SiO_2$ or $AlO_x$ resulting in TFTs with good insulation and good performance at T ≤ 150 °C [14,27]. Another approach to reduce leakage is the use of multilayered gate dielectric stacks in which $SiO_2$ layers (or other low defective materials) are employed at the dielectric/semiconductor interface. In general, these layers result in lower trap densities at the interface improving device performance and stability while imposing a higher barrier for carrier injection. Sputtered $HfO_xN_y/HfO_2/HfO_xN_y$ [41] and $HfO_2/SiO_2$ [21,42] stacks showed improved interface quality and insulation properties when compared to the respective single layer high-κ dielectrics. Hsu et al. showed excellent flexible IGZO TFTs fabricated at RT with electron beam evaporated $SiO_2/TiO_2/SiO_2$ [43]. Solution based processes can also be used for depositing high-κ dielectrics such as $HfO_2$, $ZrO_2$, and $Ta_2O_5$, permitting devices with very good performance [44,45] and multilayered stacks with these processes were shown to be promising even when considering low temperatures (T < 150 °C), as shown by Carlos et al. [46], while others have shown that this approach can even improve mechanical flexibility [47]. While masking some of the high-κ dielectrics disadvantages, these approaches can significantly decrease the effective oxide $\varepsilon_r$. Moreover, while the multicomponent and multilayer concepts in dielectrics were already demonstrated, the dielectric layer composition and the gate dielectric stack architecture need to be carefully considered to obtain the best combination of performance and reliability. This is particularly relevant when imposing low thermal budgets (T ≈ 150 °C), given that in such cases the usual benefits of high temperature annealing to improve film quality cannot be considered. These low temperatures are extremely relevant in the current scenario of flexible electronics, where aspects such as hybrid integration with temperature sensitive technologies as organics or usage of unconventional substrates as paper are considered [7,11,48]. Within this context, this work presents a study of the effect of composition in multicomponent dielectric layers composed both by sputtered $Ta_2O_5$ (a high-κ dielectric with $\varepsilon_r$ = 25) and sputtered $SiO_2$. Besides having a high dielectric permittivity and an amorphous structure, $Ta_2O_5$ can be deposited by sputtering with a good growth rate without requiring the application of very high power [14]. A low thermal budget (T ≤ 150 °C) was considering in this work, for compatibility with flexible substrates. Multilayered stacks comprising these multicomponent layers and $SiO_2$ layers were also studied and IGZO TFTs employing these dielectrics were assessed in terms of performance, stability, and reliability.

## 2. Materials and Methods

### 2.1. Device Fabrication

IGZO TFTs were fabricated with a staggered bottom gate structure on Corning glass by using standard photolithography patterning procedures, with UV patterning on a Suss MA6 aligner (SUSS MicroTec, Garching, Germany). All layers were produced by radio frequency (RF) magnetron sputtering in an AJA ATC-1300F system (AJA International Inc., North Scituate, MA, USA) without intentional substrate heating. The gate electrodes were sputtered from a Mo target in an oxygen free atmosphere with an RF power density of 3.8 W/cm$^2$ resulting in a final thickness of 60 nm. The multicomponent dielectric films were produced by co-sputtering from 2 inch ceramic targets of $Ta_2O_5$ and $SiO_2$ under an argon + oxygen atmosphere and the power applied to the $Ta_2O_5$ target was varied between 50 and 150 W, while power of 150 W to the $SiO_2$ target was kept fixed, resulting in films with different $Ta_2O_5$:$SiO_2$ contents. A substrate bias of 84 V during the dielectric depositions was used as it is known to result in denser and smoother films [21]. A dielectric film of only $Ta_2O_5$ was also produced for investigating the role of the $SiO_2$ incorporation in the high-κ dielectric stacks. The 40 nm semiconductor film was sputtered from a 2 inch multicomponent ceramic target of IGZO 2:1:2 ($In_2O_3$:$Ga_2O_3$:ZnO mol) with an RF power density of 4.9 W/cm$^2$ in an argon + oxygen atmosphere, resulting in an amorphous film with a 4:2:1 (In:Ga:Zn) atomic ratio [49]. The source and drain electrodes were sputtered in the same way as the gate dielectric. All layers were patterned by lift-off technique with the exception of the dielectric which was patterned by plasma etching in $SF_6$ atmosphere.

Resulting TFTs had a width-to-length ratio (W/L) of 320/20 (μm/μm). The devices were annealed on a hot plate at 150 °C for 1 h. A schematic of the device cross-section is shown in Figure 2a. For capacitance analysis, metal-insulator-semiconductor (MIS) devices with the same dielectric layers were produced using p-type Si wafers as substrates and Mo top contacts with areas of $1.88 \times 10^{-3}$ cm$^2$. The multicomponent dielectrics were also deposited in p-type Si wafers for Rutherford backscattering spectrometry (RBS) for compositional analysis and for spectroscopic ellipsometry (SE).

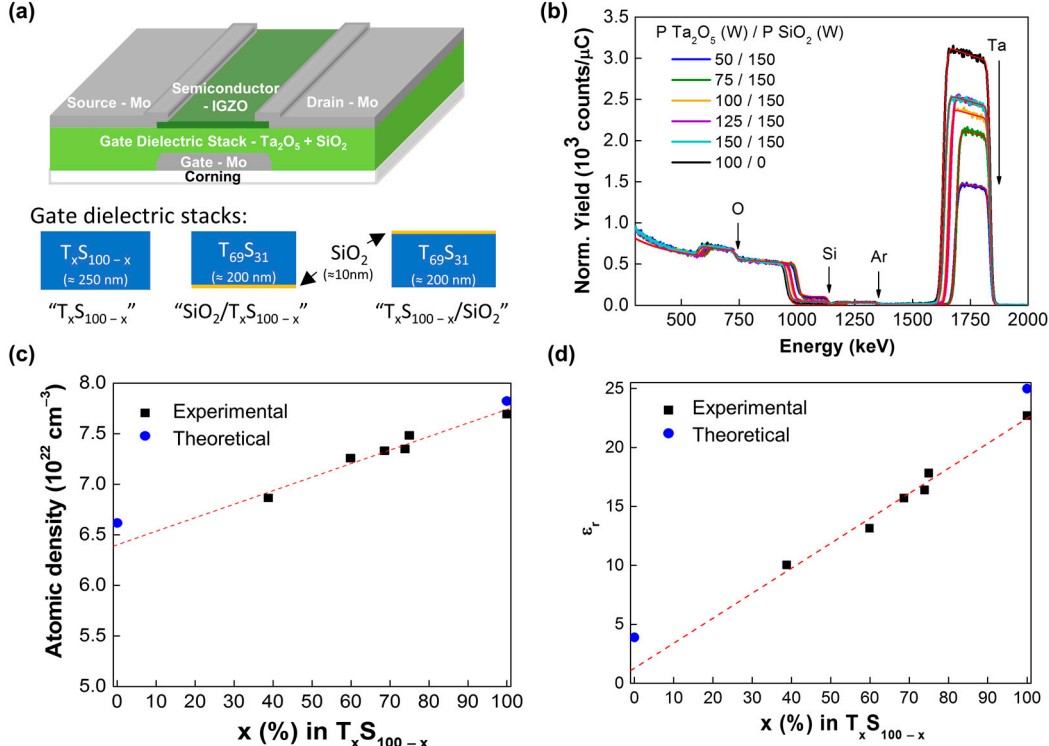

**Figure 2.** (**a**) Schematic of the device cross-section and of the single layered and multilayered gate dielectric stacks. (**b**) Rutherford backscattering spectrometry (RBS) spectra measured at 140° for films sputtered with different powers in the Ta$_2$O$_5$ target. (**c**) Atomic density of the T$_x$S$_{100-x}$ films. (**d**) Dielectric permittivity of the T$_x$S$_{100-x}$ films.

Additionally, devices employing gate dielectrics consisting of a stack of a multicomponent layers and a SiO$_2$ layer where also fabricated. In these stacks the multicomponent layer was produced with a power of 100 W in the Ta$_2$O$_5$ target and a power of 150 W in the SiO$_2$ target, with expected thickness of 200 nm for this layer. The SiO$_2$ layer in these stacks is employed either at the gate/dielectric interface or at the dielectric/semiconductor interface by sputtering with a power of 150 W before or after the multicomponent layer, respectively, and without breaking vacuum during the entire dielectric stack deposition, with a nominal thickness of 15 nm for the SiO$_2$ layer. These dielectric stacks and the multicomponent single layer dielectrics are schematized in Figure 2a.

### 2.2. Films and Devices Characterization

The stoichiometry of the dielectrics was assessed by Rutherford backscattering spectrometry (RBS) using a 2 MeV He beam delivered by a 2.5 MV van de Graaf accelerator. Two solid state detectors placed at 140° and 165° were used to collect the backscattered particles. The RBS spectra were analyzed with IBA DataFurnace NDF software [50]. A HORIBA-Jobin Yvon spectroscopic ellipsometry system was used with an incident angle of 70° in a spectral range between 1.5 and 6.5 eV. The acquired data was analyzed with DeltaPsi 2 software (v2.6.6.212, Horiba, Bensheim, Germany) and fitted using the Tauc–Lorentz dispersion formula [51]. TFTs and MISs were characterized using a Keysight

B1500A (Keysight Technologies, Santa Rosa, CA, USA) semiconductor parameter analyzer in a EPS 150 probe station (Cascade Microtech, Beaverton, OR, USA). The MIS devices were characterized through capacitance–voltage (C-V) analysis at 100 kHz.

## 3. Results and Discussion

### 3.1. Multicomponent Dielectric Properties

RBS analysis was performed on the multicomponent films sputtered with different powers applied to the $Ta_2O_5$ target. The RBS spectra are presented in Figure 2b, in which the barrier for each element is indicated by vertical arrows. It is noticeable that with the increase of the power in the $Ta_2O_5$ target the Ta barrier height increases whereas the Si barrier decreases. Analysis of this data allowed assessing of the stoichiometry of the films, confirming the different $Ta_2O_5$ and $SiO_2$ contents within the dielectrics as summarized in Table 1. RBS revealed some incorporation of Ar (values presented in Table S1) across all film compositions, as is common for films sputtered in Ar rich atmospheres [52]. Figure S1 presents the Ar content in the dielectric for different $Ta_2O_5$ contents, showing clearly that the Ar incorporation is more pronounced for higher $Ta_2O_5$ contents. Nevertheless, the Ar content across all compositions ($4.3 \pm 0.7\%$) does not change significantly (from 3.4% to 5.5%) and the $Ta_2O_5$ and $SiO_2$ contents presented in Table 1 are thus normalized to 100%, for simplicity. According to the normalized $Ta_2O_5$ and $SiO_2$ compositions the multi-component dielectric layers are denominated as "$T_xS_{100-x}$" where $x$ is the approximate $Ta_2O_5$ percentage of the material, and $100 - x$ is thus the approximate $SiO_2$ percentage. "T" and "S" thus correspond to $Ta_2O_5$ and $SiO_2$, respectively, under this nomenclature. For clarity, when the dielectric is based on a single cation, the usual chemical formulas are employed, namely, $Ta_2O_5$ or $SiO_2$. For each of these layers its thickness was extracted from the analysis of the SE data, with values in the $200-250$ nm range [53]. As for the multilayered gate dielectric stacks, their multicomponent layers have "$T_{69}S_{31}$" composition and are then denominated "$SiO_2/T_{69}S_{31}$" and "$T_{69}S_{31}/SiO_2$", according to the position of the $SiO_2$ layer in the gate dielectric stack, as schematized in Figure 2a.

**Table 1.** Normalized $Ta_2O_5$ and $SiO_2$ content in the sputtered $T_xS_{100-x}$ films, obtained by RBS, and corresponding dielectric permittivities.

| Name | Power in $Ta_2O_5$ Target (W) | $Ta_2O_5$ Content (mol.%) | $SiO_2$ Content (mol.%) | $\varepsilon_r$ |
|---|---|---|---|---|
| $T_{39}S_{61}$ | 50 | 38.8 | 61.2 | 10.0 |
| $T_{60}S_{40}$ | 75 | 59.9 | 40.1 | 13.2 |
| $T_{69}S_{31}$ | 100 | 68.6 | 31.4 | 15.7 |
| $T_{74}S_{26}$ | 125 | 73.9 | 26.1 | 16.4 |
| $T_{75}S_{25}$ | 150 | 75.0 | 25.0 | 17.8 |
| $Ta_2O_5$ | 100 | 100.0 | 0.0 | 22.7 |

Knowing the areal density extracted from RBS and the thicknesses from SE (Table S1), the atomic density for each composition can be calculated. It was shown in a previous work that at least for the range of power used here, the $Ta_2O_5$ density is independent of the power density in the $Ta_2O_5$ target [53] and since the $SiO_2$ target power density is kept for all compositions, the atomic densities for each composition can be assumed to depend linearly on the content of each material as per (1)

$$\rho = \rho_{Ta_2O_5}.x + \rho_{SiO_2}(100 - x) = (\rho_{Ta_2O_5} - \rho_{SiO_2}).x + \rho_{SiO_2} \tag{1}$$

where the approximate percentages of $Ta_2O_5$ and $SiO_2$ are $x$ and $100 - x$, respectively. Figure 2c shows the atomic density for the different dielectric compositions. From the linear fitting of the data, the oxides' atomic densities are estimated as $\rho_{SiO_2} = (6.40 \pm 0.10) \times 10^{22}$ cm$^{-3}$ and $\rho_{Ta_2O_5} = (7.74 \pm 0.17) \times 10^{22}$ cm$^{-3}$ which are below only 3.3% and 1.1%

of the theoretical values, respectively, assuming mass densities of 2.2 g·cm$^{-3}$ for SiO$_2$ [54] and 8.2 g·cm$^{-3}$ for Ta$_2$O$_5$ [55].

Regarding surface roughness, SiO$_2$ films are known to be smooth and even with the increase of Ta$_2$O$_5$ content in the multicomponent material, ellipsometry and AFM showed that these films are still smooth with T$_{69}$Si$_{31}$ layers and Ta$_2$O$_5$ films presenting roughness below 0.5 nm and 1.1 nm, respectively [53,56]. Furthermore, XRD characterization showed that these films are amorphous up to 900 °C [56].

From the C-V characterization of the MIS structures (Figure S2), the dielectric permittivity ($\varepsilon_r$) was calculated for the different multicomponent dielectrics, as presented in Figure 2d and in Table 1. As expected, by incorporating Ta$_2$O$_5$ the dielectric constant can be greatly increased when compared to that of pure SiO$_2$. Assuming that $\varepsilon_r$ depends linearly on the content of each material as per (2)

$$\varepsilon_r = \varepsilon_{r,Ta_2O_5}.x + \varepsilon_{r,SiO_2}(100-x) = (\varepsilon_{r,Ta_2O_5} - \varepsilon_{r,SiO_2}).x + \varepsilon_{r,SiO_2} \tag{2}$$

where $x$ and $100-x$ are the approximate percentages of Ta$_2$O$_5$ and SiO$_2$, respectively, each oxides' permittivity can be estimated as $\varepsilon_{r,SiO_2} = 1.29 \pm 1.06$ and $\varepsilon_{r,Ta_2O_5} = 22.48 \pm 1.82$ from the linear fitting of the data. These values are relatively lower than the theoretical permittivities (3.9 and 25, respectively) which can be attributed to the defective structure of low-temperature sputtered oxide dielectrics. This is especially noticeable for the SiO$_2$'s dielectric constant, highlighting the significance of the Ta$_2$O$_5$ incorporation as a method for achieving reasonable dielectric permittivities. Regarding the multilayered stacks, while employing SiO$_2$ layers, they in fact have a capacitance between that of the T$_{69}$S$_{31}$ and the T$_{75}$S$_{25}$ gate dielectric stacks and result in effective dielectric permittivities close to that of the T$_{69}$S$_{31}$ gate dielectric stack.

### 3.2. Device Characterization

3.2.1. TFT Performance

The performance of TFTs employing the different dielectric compositions was evaluated and a transfer curve for the T$_{60}$S$_{40}$ composition is presented in Figure 3a as an example. Figure 3b summarizes the field-effect mobilities ($\mu_{FE}$) and subthreshold slopes (*SS*) obtained with the different T$_x$S$_{100-x}$ dielectrics. With the addition of Ta$_2$O$_5$ content, a slight degradation of the mobility from $\approx$ 16.3 to 14.8 cm$^2$·V$^{-1}$·s$^{-1}$ is observed. This can be justified by the decrease of quality of the T$_x$S$_{100-x}$/IGZO interface with the addition of Ta$_2$O$_5$, where defects such as fixed charges can cause the scattering of carriers, decreasing their mobility. The trap density at the interface ($D_{it}$) was extracted from (3) in which $k$, $T$, and $e$ have their usual physical meanings and $C$ is the dielectric capacitance.

$$SS \approx ln(10)\frac{k.T}{e}\left(1 + e.\frac{D_{it}}{C}\right) \tag{3}$$

As presented in Figure 3c, $D_{it}$ tends to increase with the addition of Ta$_2$O$_5$ content, showing that the high-κ oxide results in poorer interface quality with IGZO when compared to SiO$_2$. Nevertheless, for the studied range of Ta$_2$O$_5$ incorporation, all the devices show good performance with mobilities above 14 cm$^2$·V$^{-1}$·s$^{-1}$, SS lower than 0.3 V/dec, on/off ratios close to $1 \times 10^7$ and gate leakage currents (I$_G$) close to 1 pA. Regarding the devices employing the multilayered gate-stacks, for the SiO$_2$/T$_{69}$S$_{31}$, dielectric properties close to that of the T$_{69}$S$_{31}$ were found, which should be related to the similar T$_{69}$S$_{31}$/IGZO interface. For the TFTs employing a T$_{69}$S$_{31}$/SiO$_2$ layer, mobility was slightly higher than most T$_x$S$_{100-x}$ compositions, and a great improvement is shown in SS (0.15 V/dec) and consequently $D_{it}$, showcasing the significantly better quality of the SiO$_2$/IGZO interface.

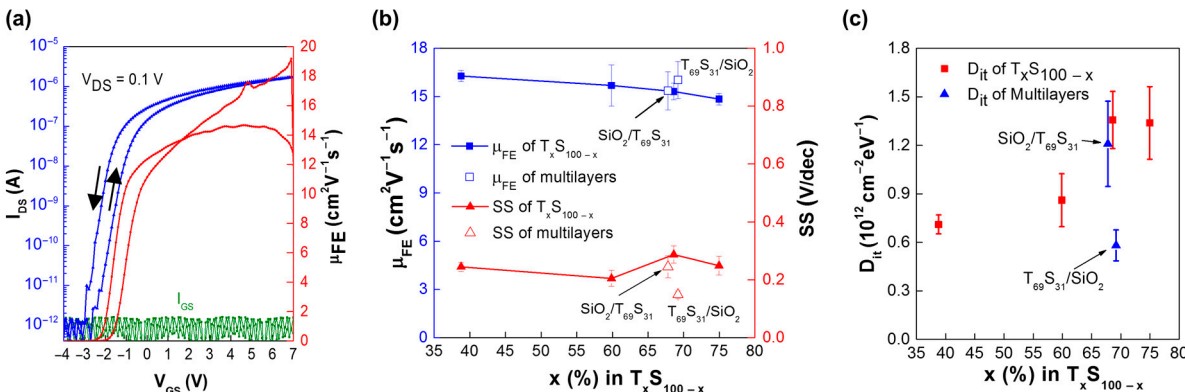

**Figure 3.** (**a**) Transfer characteristics for an In-Ga-Zn-O thin-film transistors (IGZO TFT) employing the $T_{60}S_{40}$ dielectric. (**b**) Field effect mobility and subthreshold slope and (**c**) trap density at the interface for TFTs employing the $T_xS_{100-x}$ and the multilayered dielectrics.

### 3.2.2. Insulation Reliability

Arising from higher defect densities and lower band gaps, employing high-κ dielectrics can often result in high $I_G$. In fact, TFTs employing a $Ta_2O_5$ dielectric layer revealed poor insulation which compromised the extraction of their transfer characteristics. Regarding the $T_xS_{100-x}$ dielectrics, while $I_G$ was below 1 pA for TFTs considered to be working properly (see Figure 3a), in some devices an abrupt increase of $I_G$ is seen with the increase of gate voltage. To quantify the dielectric reliability for each $T_xS_{100-x}$ composition (and for the multilayered structures), a leakage probability ($P_{Leak}$) was determined as the frequency of TFTs presenting gate leakage out of 18 similar devices. In practice, $I_G$ was either in the noise level (for properly working TFTs) or higher than several nA (considered as leakage). Figure 4a presents the leakage probability for TFTs with channel widths and lengths of 320 μm and 20 μm, respectively.

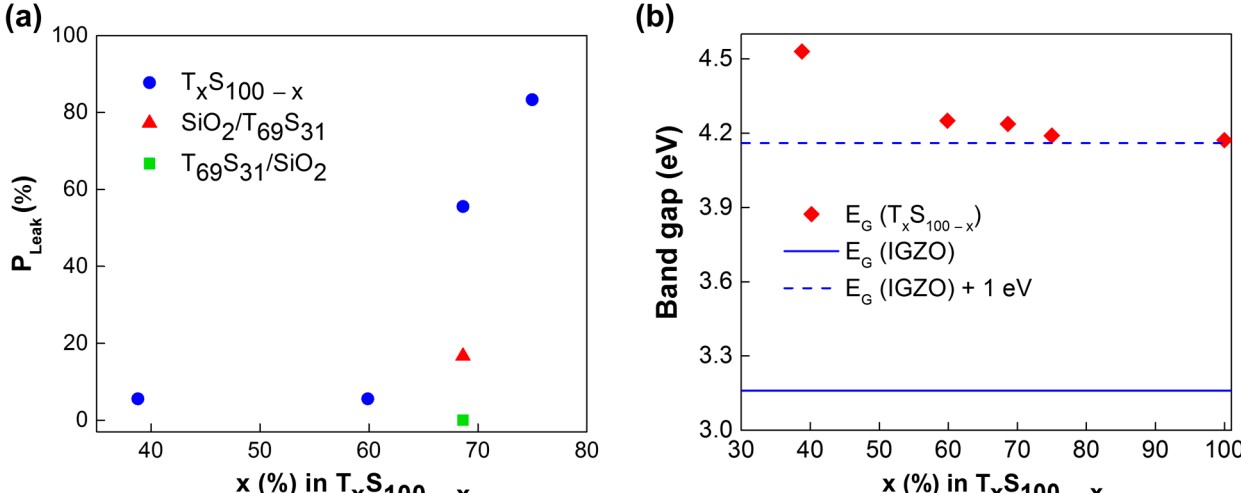

**Figure 4.** (**a**) Gate leakage probability for TFTs employing the $T_xS_{100-x}$ and the multilayered dielectrics. (**b**) Band gaps of IGZO and of the $T_xS_{100-x}$ dielectrics.

Whereas a good reliability is seen for $Ta_2O_5$ contents of 60% and lower, it decreases abruptly for contents of 69% and higher. This can be attributed either to an increase of the conductivity of the dielectric layer or to a decrease of the conduction band offset ($\Delta E_C$) between IGZO and the dielectric layer, allowing the injection of carriers into the dielectric trough conduction mechanisms such as thermionic emission and field emission [57]. The band gaps of the $T_xS_{100-x}$ dielectrics, obtained from ellipsometry measurements [53] are

shown in Figure 4b. For all compositions, the band gaps are much closer to that of $Ta_2O_5$ ($\geq 4$ eV [58,59]) than that of $SiO_2$ (8.9 eV), a trend that has been seen before for other high-$\kappa$/$SiO_2$ mixtures [14,20,35]. Nevertheless, the increase of band gap with the incorporation of $SiO_2$ is still relevant: to block charge injection, a conduction band offset above 1 eV is desirable and while the valence band offset to IGZO is unknown, the measured $T_xS_{100-x}$ band gaps are very close to 1 eV above the IGZO's band gap (as represented in Figure 4b). This means that the slight band gap increase by the incorporation of $SiO_2$ may play a critical role in blocking the injection of charge. Leakage probabilities for the multilayered dielectrics are also presented in Figure 4a, and these are compared to the $T_{69}S_{31}$ single layer. The $SiO_2$ layer at the gate/dielectric interface ($SiO_2/T_{69}S_{31}$) significantly reduces the leakage probability, demonstrating the good insulation properties of this stack. Nevertheless, with the $SiO_2$ layer at the dielectric/IGZO interface ($T_{69}S_{31}/SiO_2$), this is even further enhanced, with no leaking devices being found. Compared to the $SiO_2/T_{69}S_{31}$ layer, this improvement can be attributed to a decrease of charge injection from the IGZO to the gate dielectric stack (e.g., hot electron injection) due to either the higher $\Delta E_C$, better interface quality, or both. This shows that multilayered gate dielectric stacks are a viable approach to improve device performance without sacrificing reliability.

### 3.2.3. Stability

It is interesting to notice that the transfer curves for these devices present counterclockwise hysteresis. When employing $SiO_2$ dielectrics the hysteresis is known to be clockwise resulting from electron trapping in the dielectric/semiconductor interface trap-sites. Nevertheless, this counterclockwise hysteresis has been previously reported at times for some high-$\kappa$ dielectrics, as will be discussed later. This device hysteresis was shown to increase with decreasing $V_{GS}$ step (which increases measurement time) as shown in Figure 5a for the $T_{60}S_{40}$ composition. Considering $V_{On}$ the $V_{GS}$ for $I_{DS} \approx 100$ pA, the measured hysteresis ($\Delta V_{On}$) for different $V_{GS}$ steps for the different $T_xS_{100-x}$ compositions is presented in Figure 5b. Counterintuitively, the hysteresis is higher (in magnitude) for higher $SiO_2$ content and this will be addressed later. As for the multilayered gate dielectric stacks, the hysteresis was shown to be close to that of the corresponding single layer ($T_{69}S_{31}$), even in the case of a $SiO_2$/IGZO interface.

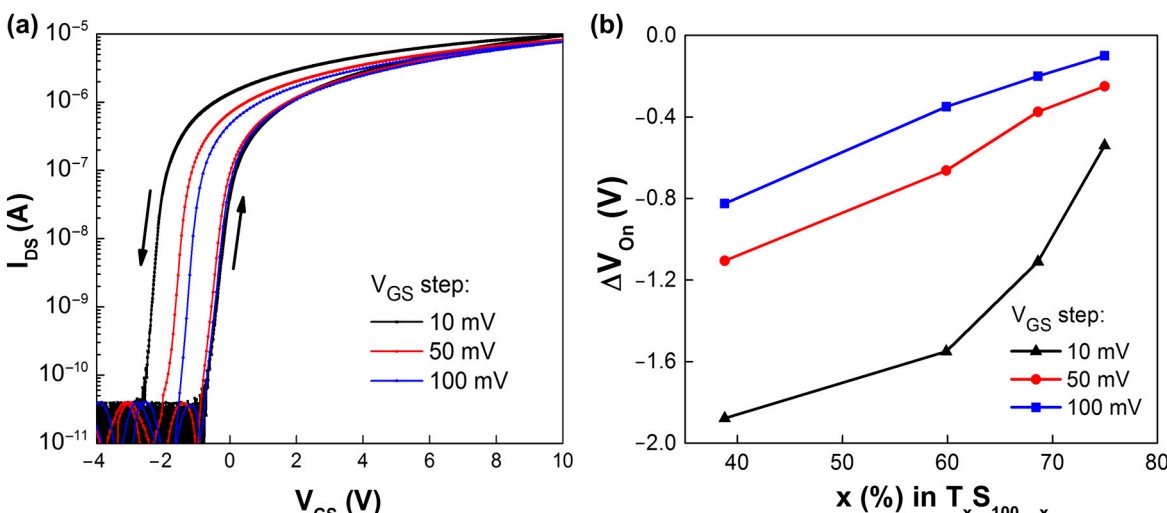

**Figure 5.** (**a**) Counterclockwise hysteresis in transfer curves with $V_{GS}$ steps of 10 mV, 50 mV, and 100 mV (for the $T_{60}S_{40}$ composition). (**b**) $\Delta V_{On}$ for the different $Ta_xSi_{100-x}$ compositions, for different $V_{GS}$ steps (lines are for eye guiding only).

The hysteresis' direction is usually tied with the direction of the $V_{th}$ shift seen during positive gate bias stress (PGBS), unless different mechanisms play a role in these [60]. TFTs with the different dielectrics where submitted to PGBS with a gate voltage of 10 V for 1 h, followed by 1 h of recovery at $V_{GS} = 0$ V. During both, a drain voltage of 0.1 V was applied

allowing addressing of the value of $V_{th}$ by (4), which was fairly consistent with the $V_{th}$ extracted from transfer curves before and after stress and recovery.

$$I_{DS} = \mu_{FE} C_{ox} \frac{W}{L} \left( (V_{GS} - V_{th}) V_{DS} - \frac{V_{DS}^2}{2} \right) \tag{4}$$

The threshold voltage shift ($\Delta V_{th}$) for the different compositions is shown in Figure 6a. Similarly to the devices' hysteresis, $V_{th}$ also shifts towards more negative values during PGBS, as opposed to the typically reported positive $V_{th}$ shift associated with electron trapping at the interface. $V_{th}$ shifts with higher magnitude are once again observed for the SiO$_2$-richer compositions, as summarized in Figure 6b. Interestingly, in the same time frame, $V_{th}$ shows only a partial recovery (for $V_{GS} = 0$ V), unlike in the faster relaxations typical of cases dominated by electron (de)trapping, whereas applying a negative gate bias ($V_{GS} = -10$ V) resulted in a fast recovery of the threshold voltage (Figure S3a).

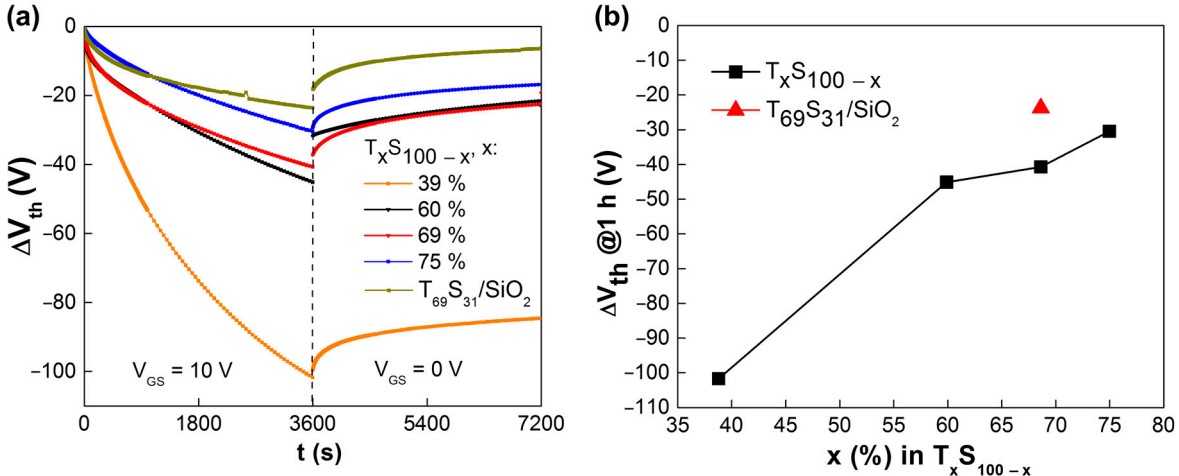

**Figure 6.** (**a**) $V_{th}$ shift during positive gate bias stress ($V_{GS} = 10$ V) and recovery ($V_{GS} = 0$ V). (**b**) $V_{th}$ shift after the 1 h stress (line is for eye guiding only).

In Figure 6b, the $V_{th}$ shift for the T$_{69}$S$_{31}$/SiO$_2$ layer is also shown. While its shift magnitude is significantly lower than that of the T$_x$S$_{100-x}$/IGZO interfaces, it is still negative, suggesting that this anomalous shift is not an interface phenomena and it is in competition with the electron-trapping at the SiO$_2$/IGZO interface during the PGBS [60].

Considering conventional applications (for which $V_{th}$ shifts are undesirable), Ta$_2$O$_5$-richer compositions present as better alternatives both by resulting in higher dielectric permittivities and for lower $V_{th}$ shifts. While these compositions can be unreliable in terms of gate leakage, it was demonstrated that thin SiO$_2$ layers at the dielectric/semiconductor interface successfully prevent gate leakage even for thinner overall gate dielectric stacks (with effectively higher capacitances). This shows that the multicomponent and multi-layered approach can be a feasible method for achieving high device performance by incorporation of high-κ dielectrics without sacrificing device reliability. In fact, while the presented $V_{th}$ shift magnitudes are considerably high, further addition of SiO$_2$ layers can be used to counterbalance this anomalous shift. This approach permits achieving devices with lower and positive $V_{th}$ shift magnitude, while still maintaining desirable values for effective $\varepsilon_r$ of 10–15. This is shown in Figure 7, for devices previously reported by our group [60] which employed similar multilayered gate dielectric stacks with an increased number of SiO$_2$ layers as shown in the inset of Figure 7b, resulting in $\varepsilon_r \approx 13$. While positive $V_{th}$ shifts in transfer curves measured after discrete periods of gate bias stress suggest the charge trapping mechanism only (Figure 7a), closer inspection along the duration of the stress by monitoring the drain current (Figure 7b) allows observing that the anomalous shift of $V_{th}$ occurs during the first few minutes of stress. This clearly shows that the two mechanisms

are in competition with charge trapping dominating, eventually resulting in an overall $V_{th}$ shift of 1.3 V after 1 h of bias stress. Similar devices were successfully applied for a flexible radiation sensing system, both for timing signals generation for addressing the sensors in the irradiated matrix [61] and for a high-gain transimpedance amplifier for amplification and voltage transduction of the sensors current signal [10], with both implementations requiring robust device operation.

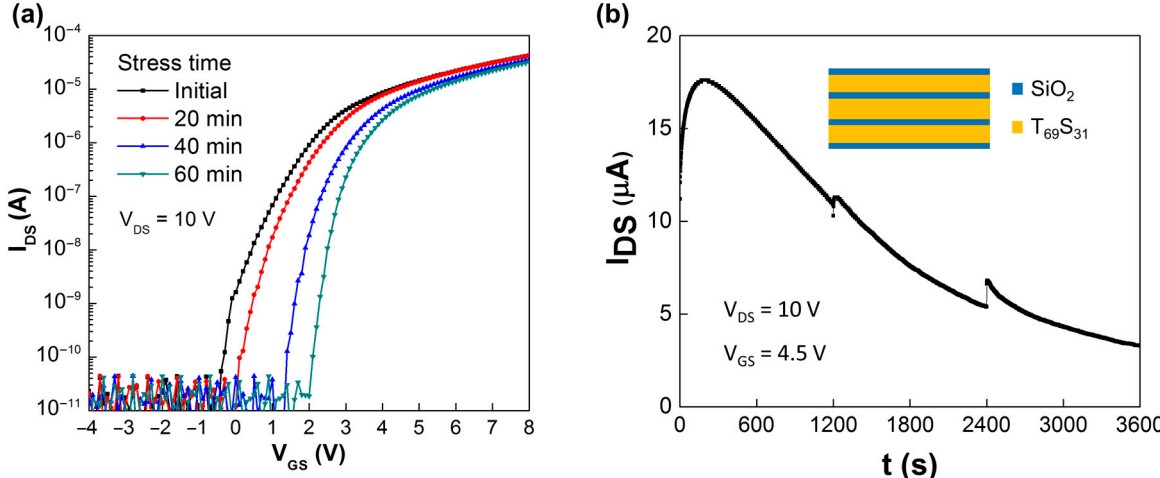

**Figure 7.** (**a**) Transfer curves and (**b**) drain current measured during positive gate bias stress (PGBS) of TFTs employing the multilayered gate dielectric stacks with increased number of $SiO_2$ layers (shown in inset). Adapted from [60].

### 3.3. Mechanism of the Anomalous $V_{th}$ Shift

A discussion of the mechanisms involved in the observed anomalous $V_{th}$ shifts is now presented. While these have been reported for TFTs employing high-κ gate dielectrics (such as $Ta_2O_5$ [62–64], $ZrO_2$ [9] and $HfO_2$ [65]), other reports often show normal shift directions. High-κ dielectrics are known to be prone to have high defect density (probably playing a role in the anomalous shift) and it should be expected that different processing methodologies can result in distinct material qualities, leading to different device behaviors. The mechanisms that are normally used to explain anomalous $V_{th}$ shifts [34] are: charge (de)trapping from the gate dielectric [62,65] ionic migration within the dielectric [63] (understandable as a slow polarization of the dielectric material [64]) and defect creation [66]. Regarding defect creation, it can lead to an increase in the carrier concentration, resulting in the decrease of the $V_{th}$, and it was shown before for poor quality semiconductors [66]. In this case, this can be disregarded as no change in the transfer curves' SS was observed, demonstrating the stability of the semiconductor (even considering the low thermal budget employed here). Furthermore, it is expected that the involved mechanism is dielectric related as this anomalous shift is not observed in our IGZO TFTs when employing other dielectrics. Regarding the charge trapping mechanisms, a positive net charge change is required for a negative $V_{th}$ shift. Hole trapping at the dielectric/semiconductor interface can be disregarded as the voltage polarity is opposite to what would be required, and wide band gap n-type oxides require optical excitation for holes to be generated in the first place. Then, electron detrapping from the dielectric to the gate must be considered (e.g., from negatively charged oxygen interstitials: $I_O^-$ or $I_O^{2-}$ [65]). However, detrapping at this interface cannot change the electron concentration at the semiconductor, and thus cannot explain the anomalous shift if charge migration is not assumed [34,64]. To investigate the mechanism, the time dependency of the threshold voltage shift during the gate bias stress was studied and a power-law dependence ($\Delta V_{th} \propto t^n$) was found, as shown in Figure 8a for the $T_{39}S_{61}$ and $T_{75}S_{25}$ compositions. While exponential dependencies are seen when charge (de)trapping is the relevant process, power dependencies are often related to reaction–diffusion mechanisms. For the compositions presented in Figure 8a, the exponent

*n* changed from ≈0.5 to ≈0.6 after approximately 1 min. For the intermediate $T_xS_{100-x}$ compositions, the exponent changed from ≈0.33 to ≈0.5 after approximately 1–5 min (Figure S4a,b). Similar dependencies were noted for the multilayered stacks (Figure S4c), further suggesting that the mechanism is not dominated by either of the interfaces. Power law time dependency of $\Delta V_{th}$ is known for the diffusion of H species in MOSFETs during negative bias temperature stress, where exponents of 1/3 and 1/2 are associated to trap-generation and diffusion of the charged species $H^+$ and $H_2^+$, respectively [67]. Aleksandrov et al. also associated *n* = 1 to first-order chemical-reaction kinetics and *n* closer to 0.5 to diffusion and drift kinetics of $H^+$ ions [68]. While further investigation is needed to understand the involved species in our devices, it is interesting to notice that $Ta_2O_5$ dielectrics are known for their ionic conductivity, with $H^+$ and $V_O$ migration in its bulk being known [69] and partly responsible for their application as resistive switching layers [70]. Regarding the $V_{th}$ recovery (for $V_{GS}$ = 0 V in Figure 6a), an exponential dependency with time is apparent: while recovering quickly initially, it seems that, at least in the presented time frame, $V_{th}$ is slowly tending to values significantly larger than the initial ones. Charge migration is often ruled out when fast recoveries are seen (as without driving force the species cannot quickly return to their initial positions), but the relatively small recovery seen across all compositions seems to imply that migration has to be considered. For further evaluation, the full recovery for the dielectric composition presenting the higher $V_{th}$ shift ($T_{39}S_{61}$) was studied and is shown in Figure 8b. The recovery took place in a much larger time frame (>1 month) than that of the stress process (1 h) and followed an exponential behavior with a time constant $\tau \approx 8 \times 10^5$ s which suggests a different recovery process than that seen up to 1 h immediately after the stress. As stated before, applying a negative $V_{GS}$ resulted in a much faster recovery of $V_{th}$ (Figure S3a), of just a few minutes. This is in agreement with negative bias stress measurements made in as-fabricated devices (Figure S3b), where a fast increase of $V_{th}$ is observed for the first minutes of stress, with the reverse $\Delta V_{th}$ direction in relation to the applied gate bias. Afterwards, $V_{th}$ is fairly stable along the stress time, given the absence of photoinduced holes (measurements under dark conditions) to sustain the commonly observed negative $V_{th}$ shifts under negative bias illumination stress (NBIS) [71].

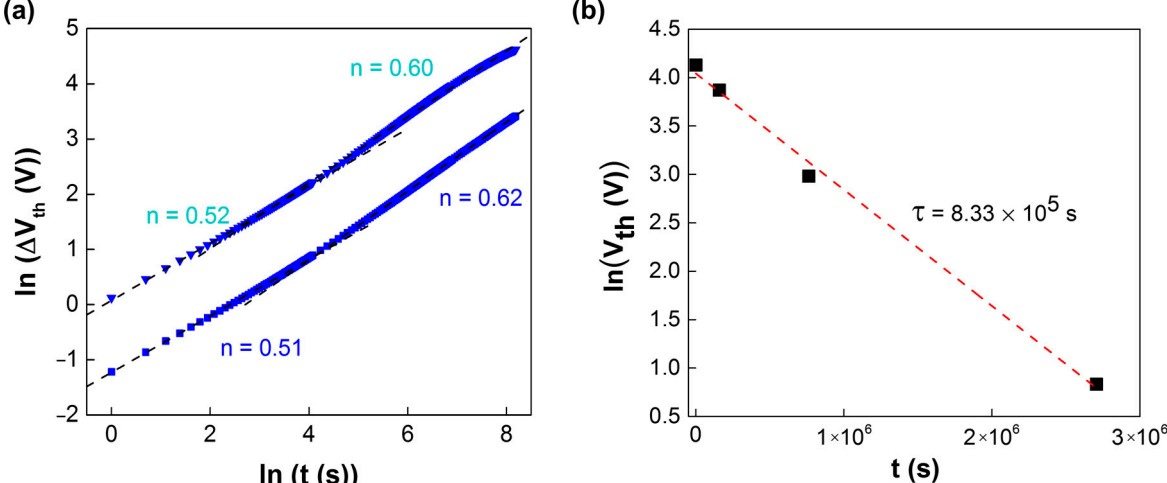

**Figure 8.** (a) $\Delta V_{th}$ during gate bias stress for the $T_{39}S_{61}$ and $T_{75}S_{25}$ compositions, showing a power law time dependency. (b) Long $V_{th}$ recovery for the $T_{39}S_{61}$ TFT, showing a very high time constant.

These results point to the anomalous shift being related to the migration of charge within the dielectric, but the nature of the migrating species or type of defects involved is not understood yet.

While $Ta_2O_5$ is known to have a considerable defect density, tending to form suboxides, the $V_{th}$ instability was found to be more pronounced for $SiO_2$-richer $T_xS_{100-x}$ compositions, implying these are more defective. Oxygen displacement in the network due to Si's higher

oxygen affinity than Ta is a possible mechanism. In fact, $SiO_2$/high-κ dielectrics interfaces are known to form defects, such as dipoles ($V_O$-$I_O$ pairs) caused by oxygen displacement (due to deferring oxygen areal densities in these materials [72,73], or oxygen vacancies [74,75]. Note, that the oxygen vacancy formation energy in $Ta_2O_5$ is low compared to other high-k dielectrics [76]. Detrapping from either these defects to the gate, their migration inside the dielectric layer, activated by gate bias, or both, can result in the observed anomalous $V_{th}$ shifts. Another possible mechanism is proton migration though the bulk of $Ta_2O_5$ [77–79], but in this case it would be expected that the anomalous $V_{th}$ shift would have to increase with increased $Ta_2O_5$ content, contrary to our findings. Further investigation should be conducted to determine the nature of the charged species involved in the anomalous $V_{th}$ shift.

## 4. Conclusions

Multicomponent gate dielectric stacks comprised of both $SiO_2$ and high-κ $Ta_2O_5$ were explored for oxide TFTs with low thermal budgets, compatible with flexible electronics (T = 150 °C). Co-sputtering of both materials as well as the addition of single $SiO_2$ layers in the gate dielectric stack were considered. While the incorporation of $Ta_2O_5$ effectively increases the relative permittivity, the resultant gate dielectric stacks present poor insulation for $Ta_2O_5$ contents ≥69%, due to either their poorer insulation capability or poorer band misalignment with the semiconductor. The multilayered approach of including a thin $SiO_2$ layer at the gate/dielectric interface improved the device reliability by decreasing the probability of devices having significant leakage current ($P_{Leak}$) from 56% to 17% when comparing devices with similar multicomponent layers ($T_{69}S_{31}$). Nevertheless, including a thin $SiO_2$ layer at the IGZO/dielectric interface not only blocks the charge injection into the dielectric, resulting in very reliable gate insulation ($P_{Leak}$ = 0%), but also results in devices with better mobility (16 $cm^2 \cdot V^{-1} \cdot s^{-1}$) and SS (0.15 $V \cdot dec^{-1}$), due to the superior interface between IGZO and $SiO_2$ than between IGZO and $T_xS_{100-x}$ mixtures (SS > 0.2 $V \cdot dec^{-1}$), with $Ta_2O_5$ richer compositions presenting the poorer interface properties ($\mu_{FE}$ = 14.8 $cm^2 \cdot V^{-1} \cdot s^{-1}$ and SS ≈ 0.25 $V \cdot dec^{-1}$ for $T_{75}S_{25}$). Counterclockwise hysteresis of the transfer curves and anomalous $V_{th}$ shifts seen during PGBS were, unexpectedly, higher in magnitude for $SiO_2$ richer compositions, with $\Delta V_{th}$ ≈ −102 V and −30 V, after 1 h stress, for the $T_{39}S_{61}$ and the $T_{75}S_{25}$ compositions, respectively. Similar behavior of devices employing IGZO/$SiO_2$ interfaces discarded the interface dominance of the processes while the power law time dependency of $V_{th}$ during gate bias stress, with exponents ≈ 0.5, points towards reaction-diffusion processes. Additionally, the recovery process was very slow, presenting a very high time constant, >8 × $10^5$ s. These results imply that the anomalous shift is caused by the migration of charged species inside the dielectric. The increase of $V_{th}$ shift magnitude with the $SiO_2$ content may suggest that it promotes oxygen vacancies in the mixture by displacing oxygen from $Ta_2O_5$ due to its higher oxygen affinity. Regardless, the anomalous shift can be effectively counterbalanced by the inclusion of additional $SiO_2$ layers in the gate dielectric stack: $\Delta V_{th}$ (at 1 h of PGBS) for the $T_{69}S_{31}$ and $T_{69}S_{31}$/$SiO_2$ stacks are ≈ −41 V and −24 V, respectively, while for the stack with increased number of $SiO_2$ layers a positive shift of only 1.3 V is obtained. This multilayer approach enables devices that are both reliable and present good performance (mobility = 16 $cm^2 \cdot V^{-1} \cdot s^{-1}$, $\varepsilon_r$ ≈ 13) and that can be successfully implemented into novel flexible electronic applications.

**Supplementary Materials:** The following are available online at https://www.mdpi.com/2673-3 978/2/1/1/s1, Figure S1. Ar concentration as a function of the $Ta_2O_5$ concentration, Figure S2. C-V curves of the MIS employing the $T_xS_{100-x}$ dielectric layers. The composition and geometrical properties of the MIS are presented in the table, Figure S3. (a) $V_{th}$ shift measured during gate biasing with $V_{GS}$ = 10 V, $V_{GS}$ = 0 V and $V_{GS}$ = −10 V, sequentially, for the $T_{69}S_{31}$ composition. (b) $V_{th}$ shift measured during negative gate biasing with $V_{GS}$ = −10 V for an as-fabricated device with the $T_{39}S_{61}$ composition, Figure S4. $\Delta V_{th}$ during positive gate bias stress ($V_{GS}$ = 10 V) for the (a) $T_{60}S_{40}$, (b) $T_{69}S_{31}$ and (c) $T_{69}S_{31}/SiO_2$ compositions, showing power law time dependencies, Table S1. Molar concentrations and film density obtained by RBS analysis of $Ta_2O_5$, $SiO_2$ and Ar in the thin films deposited using different powers in the $Ta_2O_5$ and $SiO_2$ targets. Thickness of the films extracted by SE from [53].

**Author Contributions:** Conceptualization, P.B., J.V.P., and J.M.; methodology, A.R., E.A., J.M., J.V.P., and T.G., validation, J.M., J.D., and P.B.; formal analysis, J.M. and J.V.P.; writing—original draft preparation, A.R., J.M., and J.V.P.; writing—review and editing, J.M., A.K., J.V.P., A.R., T.G., J.D., E.A., R.M., E.F., P.B.; supervision, P.B. and A.K.; funding acquisition, A.K., P.B., E.F., and R.M. All authors have read and agreed to the published version of the manuscript.

**Funding:** This work is funded by FEDER funds through the COMPETE 2020 Programme and National Funds through the FCT—Fundação para a Ciência e a Tecnologia, I.P., under the scope of the projects UIDB/50025/2020, PTDC/NAN-MAT/30812/2017, and the doctoral grant SFRH/BD/ 122286/2016. This work also received funding from FEDER funds through the PORLisboa 2020 and PORAlentejo 2020 Programme, project 17852 (ORABAC). This work also received funding from the European Community's H2020 program under grant agreement No. 716510 (ERC-2016-StG TREND) and No. 787410 (ERC-2018-AdG DIGISMART).

**Institutional Review Board Statement:** Not applicable.

**Informed Consent Statement:** Not applicable.

**Data Availability Statement:** The data presented in this study are available on request from the corresponding authors.

**Conflicts of Interest:** The authors declare no conflict of interest. The funders had no role in the design of the study; in the collection, analyses, or interpretation of data; in the writing of the manuscript, or in the decision to publish the results.

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
