# Peer review of "Ta2O5/SiO2 Multicomponent Dielectrics for Amorphous Oxide TFTs"

_electronicmat, doi:10.3390/electronicmat2010001_

Round 1

Reviewer 1 Report

This paper describes the use of SiO2, Ta2O5 and mixed (Ta,Si)Ox films as dielectrics for amorphous IGZO based field-effect transistor devices. The dielectric films were prepared by sputtering and characterized by Rutherford backscattering and capacitance voltage characterization. Films were used in transistor devices and their output and transfer characteristics were evaluated, showing field effect mobilities of up to 16 cm2/Vs and on/off ratios of up to 10^7 for operating voltages of less than 10 V.

High-k dielectric materials are known to exhibit hysteresis. The analysis of shifts in threshold voltage under bias is thorough and show that the approach of using thin interfacial SiO2 layers can mitigate this problem. This work shows that many of the undesirable characteristics of high k dielectrics like Ta2O5 can be mitigated by placing thin passivating layers of SiO2 between the high k dielectric and other layers. The paper is well written and suitable for publication in electronic materials pending minor revisions:

  • Dielectric constants should be added to Table 1.
  • The TXS100-X naming system doesn’t match the oxides against which they are compared (Ta2O5 and SiO2)
  • The authors should cite the following studies of Ta based dielectrics in the introduction:

(J Heo et al Advanced Functional Materials 28 (28), 1704215) and (SY Park et al Journal of Materials Chemistry C 7 (15), 4559-4566)

Reviewer 2 Report

In this manuscript, the authors reported dielectrics based on sputtered TaOx and SiOx. Dielectrics with different composition and structures are investigated. In the main text, there are many interesting observations and results, which deserve further investigation. It is recommended to be published in Electronic Materials if the following comments can be properly addressed.

  1. In the abstract, it is mentioned that: “A stack comprising a TaxSiO100-xOy with x = 69 %”. The “%” should be deleted.
  2. In the introduction, it is mentioned that: “Ta2O5 was combined with SiO2 or AlOx resulting in TFTs with good insulation and good performance at T ≤ 150 °C”, which indicates that similar work has been published before. Can the author indicate what is the major difference between the current research compared to the previous report?
  3. In Materials and Methods, it is indicated that: “The 40 nm semiconductor film was sputtered from a 2 inch multicomponent ceramic target of IGZO 2:1:2 (In2O3:Ga2O3:ZnO mol) with an RF power density of 4.9 W/cm2 in an argon+oxygen atmosphere, resulting in an amorphous film with a 4:2:1 (In:Ga:Zn) atomic ratio”. Why in the target the atomic ratio of In:Ga:Zn is 4:2:2, while in the film, it becomes 4:2:1?
  4. It is suggested to restructure the device fabrication part to make all the fabrication processes in the right sequence of the device fabrication. Moreover, the device geometry is suggested to be added.
  5. In the manuscript, the author used “TxSy” to present TaxSiy, it is suggested to avoid using T and S to present Ta and Si respectively, as this would lead to confusion.
  6. Can the Hydrogen content be detected by RBS?
  7. Please provide the original data/figure of AFM and C-V characterization.
  8. Please provide the J-V curve of the MIS capacitors.
  9. Please provide the leakage current in the transfer characteristics.
  10. It is not clear what does “leakage probability” mean.
  11. In Figure 6, a Vth shift of up to about 100 V is obtained, which is abnormal for a bias voltage of only 10 V. Please provide the original transfer curve to extract the Vth.
  12. In Page 9, it is mentioned that: “This is shown in Figure 7, for devices previously reported by our group [57] which employed similar multilayered gate dielectric stacks with an increased number of SiO2 layers.” Please specify the “similar multilayered gate dielectric stacks”
  13. It is mentioned that mobile Hydrogen could be the reason for the counterclockwise hysteresis and Vth shifts. There are many new references reporting such phenomenon, such as “J. Am. Chem. Soc.2020, 142, 28, 12440–12452”, “Chem. Sci., 2016,7, 6337-6346”. Please revise the discussion along with these references.

Reviewer 3 Report

  1. Lines 83-88: This is the end of the state of the art, but there is not conclusion why the Ta2O5 is chosen for preparing three types of gate dielectrics.
  2. Lines 89-93: Please rewrite the objectives of the work showing the novelty of the approach to the theme.
  3. Please add one graph of the XRD showing the amorphous structure of the material.
  4. Conclusions: three types of gate dielectric layers needs to be compared numerically to show the best solution.
